# Ultrastructural Characterization of Human Bronchial Epithelial Cells during SARS-CoV-2 Infection: Morphological Comparison of Wild-Type and CFTR-Modified Cells

**DOI:** 10.3390/ijms23179724

**Published:** 2022-08-27

**Authors:** Flavia Merigo, Virginia Lotti, Paolo Bernardi, Anita Conti, Andrea Di Clemente, Marco Ligozzi, Anna Lagni, Claudio Sorio, Andrea Sbarbati, Davide Gibellini

**Affiliations:** 1Anatomy and Histology Section, Department of Neurosciences, Biomedicine and Movement Sciences, University of Verona, 37134 Verona, Italy; 2Microbiology Section, Department of Diagnostic and Public Health, University of Verona, 37134 Verona, Italy; 3General Pathology Section, Department of Medicine, University of Verona, 37134 Verona, Italy

**Keywords:** SARS-CoV-2 virus, transmission electron microscopy (TEM), double-membrane vesicles (DMVs), cystic fibrosis, CFTR, ACE2

## Abstract

SARS-CoV-2 replicates in host cell cytoplasm. People with cystic fibrosis, considered at risk of developing severe symptoms of COVID-19, instead, tend to show mild symptoms. We, thus, analyzed at the ultrastructural level the morphological effects of SARS-CoV-2 infection on wild-type (WT) and F508del (ΔF) CFTR-expressing CFBE41o- cells at early and late time points post infection. We also investigated ACE2 expression through immune-electron microscopy. At early times of infection, WT cells exhibited double-membrane vesicles, representing typical replicative structures, with granular and vesicular content, while at late time points, they contained vesicles with viral particles. ∆F cells exhibited double-membrane vesicles with an irregular shape and degenerative changes and at late time of infection, showed vesicles containing viruses lacking a regular structure and a well-organized distribution. ACE2 was expressed at the plasma membrane and present in the cytoplasm only at early times in WT, while it persisted even at late times of infection in ΔF cells. The autophagosome content also differed between the cells: in WT cells, it comprised vesicles associated with virus-containing structures, while in ΔF cells, it comprised ingested material for lysosomal digestion. Our data suggest that CFTR-modified cells infected with SARS-CoV-2 have impaired organization of normo-conformed replicative structures.

## 1. Introduction

Coronaviruses are enveloped viruses with a single-stranded positive-sense RNA genome that infect birds, mammals and humans, causing acute and persistent infections [1,2]. Seven coronaviruses induce diseases in humans: SARS-CoV [3], MERS-CoV [4] and SARS-CoV-2 [5]. They elicit a severe respiratory infection with a high mortality rate [6,7,8], whereas HCoV-OC43 [9], HCoV-229E [10,11], HCoV-NL63 [12] and HCoV-HKU1 [13] are involved in upper respiratory airway infections with low clinical consequences, with the exception of some rare cases of severe manifestations, most commonly observed in elderly, young and immunocompromised patients [7,14]. SARS-CoV-2 belongs to the *β*-*coronavirus* genus and is the causative agent of COVID-19 and, therefore, its global pandemic [15]. Like those of the other coronaviruses, SARS-CoV-2 virions are spherical with a diameter of 60–140 nm, although they display a relatively high degree of pleomorphism compared to other coronaviruses [5]. The replication cycle of SARS-CoV-2 follows the classical picture of coronavirus replication that takes place in the cytoplasm. SARS-CoV-2 entry into target cells is mediated by interaction between the receptor-binding domain of the SARS-CoV-2 Spike (S) protein and the cell surface protein angiotensin-converting enzyme 2 (ACE2), the major cellular receptor for SARS-CoV-2 infection [16,17,18]. It is noteworthy that transmembrane serine protease 2 (TMPRSS2) activity plays an important role in SARS-CoV-2 infection through S protein cleavage and the activation of the S2 domain and the subsequent fusion of viral envelope with the host cell membrane [16,18,19]. The budding of the nucleocapsid into the membrane containing the viral structural proteins allows the formation of SARS-CoV-2 virions to occur, which remain within vesicles and are not free in the cytoplasm [20]. In more detail, following host cell entry, coronaviruses’ positive-sense genomic RNA strand is recognized as mRNA by ribosomes, which, in turn, translate the viral proteins, initiating the replication–transcription complex at the endoplasmic reticulum (ER) [21]. Thus, a remodeling of intracellular membranes starts to form the replication membranous web (RMW), a three-dimensional structure, within which virus replication occurs very productively due to the proximity of all necessary factors and the capacity of the RMW to hide viral RNA from cytoplasmic innate immunity sensors [22,23]. Morphologically, the RMW is composed of multiple interconnected single- or double-membrane vesicles, called double-membrane vesicles (DMVs) and convoluted membranes (CMs). DMVs are round or oval formations of approximately 200–300 nm that are likely formed from the ER by non-structural viral proteins [21,24]. Here, after infection, viral RNA replication begins [25]. CMs can appear in the early stages after infection in association with DMVs and the ER as single-membrane reticular structures measuring 200–600 nm. These structures play an important role in DMV formation [23] and can be used as storage for non-structural proteins excessed from DMV formation [22]. Some observations of coronavirus replication cycle performed using electron microscopy, in particular, transmission electron microscopy (TEM), have described the morphology and morphogenesis of SARS-CoV [26,27], MERS-CoV [28,29,30] and SARS-CoV-2 [17,31,32,33]. A recent study [34] demonstrated that SARS-CoV-2 replication is reduced in bronchial cells from people with cystic fibrosis (pwCF), a lethal genetic disease with autosomal recessive inheritance caused by mutations in the *CFTR* gene [35], which codes for the CFTR membrane protein, an ion channel involved in both chloride and bicarbonate ion transport [36,37,38,39]. Mutations decrease the amount and/or function of CFTR, leading to abnormal ionic conductance and viscous secretion in lungs and other organs [40,41]. Due to their pre-existing pulmonary disease, pwCF may be considered at high risk of developing severe symptoms of COVID-19 [42]; however, the data from Lotti and co-workers [34] together with some epidemiological studies [42,43,44] indicated a counterintuitive situation: CFTR protein can play a major role in the regulation of SARS-CoV-2 replication machinery upon its entry into the host cell, possibly due to ionic dysregulation caused by CFTR loss of function. Noteworthily, to our knowledge, no study on SARS-CoV-2 morphogenesis in CFTR-modified cells is currently available.

To shed light on the peculiar defects in replication efficiency of SARS-CoV-2 in CFTR-modified cells, we performed an array of experiments to compare the SARS-CoV-2 replication cycle in bronchial cells with or without CFTR modification using an ultrastructural approach.

## 2. Results

### 2.1. Ultrastructural Morphology of Mock-Infected and SARS-CoV-2-Infected CFBE41o- Wild-Type Cells

First, we studied the ultrastructural morphology of mock-infected CFBE41o- wild-type (WT) cells, which showed a sparse electron-dense cytoplasm, a large nucleus with diffuse chromatin and one well-evident nucleolus (Figure 1A). In the perinuclear area, two or three small Golgi complexes were detected. ER was poorly represented and some-times showed dilated cisternae, which were scattered in the cytoplasm without any particular association with the Golgi complex (Figure 1B). The mitochondria displayed a regular morphology and an elongated shape with a homogeneous matrix and normo-conformed ridges (Figure 1C). The presence of irregularly shaped vacuoles was sometimes noted; this was likely related to physiological senescence phenomenon of some cultured cells (Figure 1D).

We also studied CFBE41o- WT cells infected with SARS-CoV-2. At early stages (1–6 h post infection (hpi)), the Golgi apparatus, ER and perinuclear cisternae appeared moderately dilated and with electron-dense contents. The cytoplasm contained irregularly shaped mitochondria and numerous polyribosomes that were distributed on the surface of the cisternae (Figure 1E). Homogeneous and electron-dense secretory granules were found scattered in the cytoplasm; we considered these to be proteinaceous in nature (Figure 1F,G). In apposition to the nuclear membrane, several profiles of DMVs were visible (Figure 1F); these structures were not observed in mock-infected cells. The DMVs were approximately 100–300 nm in diameter, usually had a rounded or oval shape and sometimes appeared completely enveloped by a double membrane, although the membrane was apparently interrupted in some tracts (Figure 1G,H). The outer and inner DMV membranes were tightly packed and in continuity with the ER membranes or the surrounding structures (Figure 1H,I). Their contents were of a homogeneously diffuse and granular matrix with an electron density similar to that of the cytoplasm. Nonetheless, during the course of infection, different profiles of DMVs were identified based on their content: some appeared to be filled with a granular dense matrix, as described above, while others contained vesicles that were externally surrounded by a coating and embedded in the matrix (Figure 1J,K). The vesicular profiles were relatively homogeneous in size and, thus, considered to be viral particles assembling.

At late stages of infection (24–72 hpi), CFBE41o- WT cells exhibited polymorphic structures that were variable in size and complexity and consisted of both elongated and dilated cisternae, likely originating from the Golgi apparatus and ER, respectively (Figure 2A). These structures often contained small, round, electron-dense particles, possibly of viral nature. DMVs were observed to be non-uniform in shape and size, but well distributed throughout the cytoplasm, with the same appearance as described above (Figure 2B–D). Moreover, especially in the latest periods of infection that we assessed (48–72 hpi), most vesicles appeared to be closed and completely filled with viral particles that were tightly packed together, showing, at high magnification, the dot-like densities of the core (Figure 2E–J). Viral particles were present both in vesicles with a scant, granular content (Figure 2K) and in matrix-free vesicles with an electron-lucent appearance that were located close to the cellular membrane and probably representing a late stage of the virus assembly. At 72 hpi, mature virions were detected on the plasma membrane as well as on the surface of microvilli and in the space between cells (Figure 2L,M). DMVs similar in size, structure and content were also observed in senescent cells at different time points after infection, suggesting the capability of these cells to release viral particles (Figure 2N,O). At this time point of analysis, proteinaceous material was still present, although the amount was reduced compared to its level in the early hours post infection.

Moreover, from 48 hpi onward, the peripheral cytoplasm of many WT cells contained electron-dense and double-membrane-bound structures, which appeared to be autophagosomes; these were always observed in cells in which the replicative structures previously described were clearly recognizable (Figure 3A–G). The autophagosomes were large and highly irregular in shape, ranging from 480 to 1600 nm in size, and were especially evident at the latest time point of analysis (72 hpi). Their content was polymorphic in appearance, mainly consisting of vesicular structures and resembling the DMVs with granular content observed at early times post infection (Figure 3D,G–J); some also contained viral particles with well-evident nucleocapsids (Figure 3D,K–M).

### 2.2. Ultrastructural Morphology of Mock-Infected and SARS-CoV-2-Infected CFBE41o- ΔF Cells

The ultrastructural morphology by TEM highlighted that mock-infected CFBE41o- ΔF cells exhibited cytoplasm with a small Golgi complex, normal polyribosomes and short ER strips (Figure 4A,B). The mitochondria were round or elongated, but smaller in both number and size in comparison to those in WT cells (Figure 4C). Most of these cells exhibited a more consistent ER dilation and vesiculation compared to WT cells.

We also studied CFBE41o- ΔF cells infected with SARS-CoV-2. In the early stage of infection (1–6 hpi), in many CFBE41o- ΔF cells, dilations of ER became increasingly pronounced over time, showing many irregularly shaped vesicles with a scant material content or with an electron-lucent appearance (Figure 4D). The proteinaceous electron-dense granules previously described were also scattered in the cytoplasm of ΔF cells, but to a lesser extent than in WT cells (Figure 4E). In comparison to WT cells, only a few ΔF cells showed DMV formations, which were variable in shape but had a tendency to be less rounded (Figure 4E–H). Some DMVs appeared limited by a double membrane with intervening clear space, in contrast to the tightly packed membranes observed in WT cells, while others were enveloped by a single membrane (Figure 4I). Moreover, other DMVs displayed degenerative figures, such as membranous lamellae in an onionskin pattern (myelin figures), which appeared in continuity with the DMV membranes (Figure 4I,J).

In the late stage of infection (24–72 hpi), DMVs observed in ΔF cells maintained the irregular shape described at early time points (Figure 5A–C), but exhibited a predominantly granular content, typical of the early time points of WT cells (Figure 5D–F), indicating a delay in the formation time of these structures. DMVs also appeared to be filled with vesicles, as observed in WT cells, but from 48 hpi onward, rather than from 24 hpi, as it was observed in WT cells (Figure 5G–I). In addition, at 72 hpi, electron-lucent vesicles with a distinct limiting membrane, like those highlighted in WT cells, were observed. Their content appeared much less organized compared to that of WT cells and consisted of viral particles that were not clearly distinguishable from each other because they appeared in aggregates, lacking well-organized distribution (Figure 5J–O). Moreover, the nucleocapsid, with a dot-like electron density characteristic, was not observable inside the viral particles.

At early time points of infection, in contrast to the observations in WT cells, autophagosomes with an irregular shape were highlighted throughout the cytoplasm of ΔF cells. These autophagosomes contained cytoplasmic structures, such as poorly preserved mitochondria and lipid droplets, as well as an electron-dense structureless content (Figure 6A–C). At late time points of infection, these autophagosome structures were still present and showed a cytoplasmic matrix content (Figure 6D). However, autophagosomes showing ER membranes concentrically arranged or vesicles with a highly electron-dense content were observed mainly at 72 hpi (Figure 6E–G).

In some ΔF cells, we also observed two or three lipid droplets per cell distributed throughout the cytoplasm. These droplets had a smooth round outline surrounded by an osmiophilic ring and a moderate electron density and were small at early (Figure 6H–J) and larger at late (Figure 6K,L) times of infection, particularly at 72 hpi.

### 2.3. ACE2 Immunoelectron Microscopic Localization in Mock-Infected and SARS-CoV-2-Infected CFBE41o- Wild-Type Cells

In mock-infected WT cells, ACE2 expression, visible as black, electron-opaque, DAB deposit, was localized on short portions of the plasma membrane, on vesicles and ER cisternae (Figure 7A). ACE2 staining was much more evident in SARS-CoV-2-infected cells at 24 hpi, when the immunoreactivity was concentrated on the cell membrane and on membrane of cytoplasmic structures, such as dilated ER cisternae, curved profiles of ER and DMVs (Figure 7B,C). In DMVs, the labeling was visible on both the inner and outer membrane and was sometimes more pronounced in specific tracts (Figure 7D–F). A similar ACE2 distribution was observed at 48 and 72 hpi, when the immunoreactivity was predominantly localized on virion-containing vesicles. Even though cells at this time point of infection showed an evident presence of viral particles in their interior, they did not exhibit any morphological changes (Figure 7G). At 72 hpi, the dense cytoplasm did not show ACE2 immunoreactivity; rather, it was found to be mainly associated with virion-containing vesicles (Figure 7H–J). The membrane of some vesicles was clearly detected as immunoreactive and dark virions were also observed free in the cytoplasm (Figure 7K), as well as on the surface of the cell membrane (Figure 7H,I), which showed a discontinuous ACE2 immunoreactivity (Figure 7L).

### 2.4. ACE2 Immuno-Electron Microscopic Localization in Mock-Infected and SARS-CoV-2-Infected CFBE41o- ΔF Cells

In mock-infected ΔF cells, ACE2 immunolabeling, evaluable as black, electron-opaque, DAB deposit, was consistent with that observed in mock-infected WT cells, demonstrating positivity mainly on membranes of the cell surface and ER (Figure 8A). Similarly, in SARS-CoV-2-infected cells at 24 hpi, ACE2 immunoreactivity was observed on both dilated ER membranes (Figure 8B,C) and the cell membrane, with a punctate pattern (Figure 8D,E). From 24 hpi, but especially at 48 and 72 hpi, DMVs with a granular content were wrapped by ACE2-immunoreactive membranes, some of which appeared to be surrounded by a single membrane (Figure 8F,I–K). Other DMVs appeared to be limited by a double membrane with intervening clear space and ACE immunoreactivity was limited to certain traits of the cell membrane (Figure 8G,H,L). Moreover, in contrast to WT cells, at 72 hpi, ΔF still showed ER membranes that were ACE2 positive in some tracts (Figure 8J). In comparison to WT cells, ΔF cells exhibited cytoplasmic vesicles that appeared empty (Figure 8M) or containing few viral particles (<10), similar in size to those observed in WT cells but lacking ACE2 immunoreactivity (Figure 8N). Few vesicles showed ACE2-immunoreactive virions and those that did contained one to three per vesicle (Figure 8O). The cell membrane surface appeared to be discontinuously ACE2 immunoreactive (Figure 8O).

All reported data are summarized in Table 1.

## 3. Discussion

Previous analyses of ultrathin sections using TEM described how coronaviruses use host cell membranes to form specialized structures, such as DMVs and CM, during their replication cycle, a process already described in a wide range of cultured cells or tissues [45]. However, the morphology and the process of development of these structures is still not well elucidated, especially for SARS-CoV-2, the most recently emerging human coronavirus. Moreover, to our knowledge, no study on SARS-CoV-2 morphogenesis in CFTR-modified cells or biopsies from pwCF is currently available.

In this report, the ultrastructural morphological changes occurring throughout the SARS-CoV-2 infection cycles in WT and ΔF CFBE41o- cells were compared using TEM.

We first described the appearances of the DMVs in the cytoplasm of WT SARS-CoV-2-infected cells, which did not appear in mock-infected cells and changed and increased in number throughout the course of SARS-CoV-2 infection. At an early stage of infection (1–6 hpi), DMVs with different content patterns were identified: (i) vesicles partially or completely filled with a granular and cytoplasmic content that resembled those previously detected in SARS-CoV-infected Vero cells at 3 and 6 hpi [25,46] and (ii) vesicles containing circular profiles lying in the matrix, identified as viral particles based on their size and location. DMVs with such characteristics could represent intermediate forms of a not-yet-completed viral replication process. DMVs of both types identified were communicating with the surrounding cytoplasm through either the continuity of their membranes with those of the ER or partial interruptions in their membrane. Due to the assumption that DMVs are closed structures, this finding was unexpected [23,47]. Only recently, Wolff et al. [48] demonstrated by cryo-electron microscopy the presence of a molecular pore in DMV membranes of coronavirus-infected cells, making these structures an attractive site for viral replication. Although the intracellular compartment in which SARS-CoV-2 replication and budding take place has not yet been fully clarified, previous studies indicate that DMVs are the site of viral genome replication [24,49]. Moreover, immunofluorescence analysis revealed localization of double-stranded RNA, an intermediate for viral RNA, in DMVs of SARS-CoV-infected Vero cells [50], and non-structural proteins have been reported to colocalize with newly synthesized RNA in SARS-CoV-induced DMVs [51,52]. In addition, metabolic labeling of newly synthesized viral RNA in SARS-CoV-2-infected cells suggested that DMVs are a primary site of viral RNA synthesis and neither ER nor other double-membrane structures seemed to play a role in this process [25]. On these bases, it is possible to assume that the heterogeneous DMV phenotypes observed in this study may represent transitional stages of the SARS-CoV-2 replication process, evolving over the course of infection.

At the latest stages of infection we studied (24–72 hpi), vesicular profiles that were quite different from those previously described were observed in SARS-CoV-2-infected WT cells as isolated structures, both membrane and non-membrane bound, that contained a variable number of viral particles that were clearly distinguishable as mature virions due to their morphologic features, such as size and the dot-like electron densities of the nucleocapsid [53]. In appearance, these structures were identical to those previously described in a variety of human tissues or cultured cells at late stages of viral infection, especially at 48–72 hpi [49]. Interestingly, some individually encapsulated, formed virions were clearly visible on the cell membrane after being released outside the host cell by exocytosis. Since all these characteristics are hallmarks of late stages of the viral cycle, we speculate that WT cells can reproduce the asynchronous nature of SARS-CoV-2 infection.

In contrast, we observed that only a few SARS-CoV-2-infected ΔF cells exhibited DMVs with morphological characteristics, like those observed in WT cells at early stages of infection. Some DMVs showed an irregular shape and even degenerative changes, such as the presence of myelin figures, tightly packed membranous whorls with the peripheral membrane in continuity with those of DMVs. Vesicles with the typical characteristics of mature forms were observed in smaller quantities compared to WT cells only at 72 hpi, with a viral content lacking a well-organized distribution and structure. These observations are in accordance with the finding of our recent study, reporting a higher viral copy number in WT compared to ΔF cells, possibly due to a defect in viral replication linked to defective expression/function of CFTR protein [34]. Similarly, virions were frequently observed at the cell surface in WT and only occasionally detected in ΔF cells, indicating differences in SARS-CoV-2 replication and release between the two cell strains.

It is noteworthy that we observed peculiar morphological alterations after SARS-CoV-2 infection: autophagosomes were observed in both CFBE41o- cell lines analyzed, while lipid droplets were detected only in SARS-CoV-2-infected ΔF cells, starting from an early stage of infection.

In viral infections, autophagy can act by directly degrading the viral components, by regulating the inflammatory response or by activating the downstream signaling pathway that induces autophagosome formation [54]. Recent evidence has proven that various viruses have evolved different strategies to co-opt autophagy in favor of their own life cycle [55]. Several positive-stranded RNA viruses, for example, activate autophagy to generate autophagosomes, which are accumulated by blocking lysosomal fusion and used as structures for viral replication [56,57]. Moreover, recent studies supported this observation, showing that SARS-CoV-2 also subverts autophagy to accelerate its replication [58,59,60]. In our study, we observed the presence of autophagosomes in both cell lines but with a different time of onset: they appeared late in infection (48–72 hpi) in WT and early (from 3 hpi onward) in ΔF cells. In addition, the autophagosome patterns were remarkably different in the two cell lines: In WT cells, the autophagosomes were mainly composed of vesicles apparently similar to DMVs and, therefore, strictly associated with virus-containing structures, while in ΔF cells, the material sequestered inside the autophagosomes consisted of the typical ingested material destined for lysosomal digestion, being mainly composed of cytoplasmic organelles, such as mitochondria and inclusions, such as lipid droplets. Based on these data, we can assume that in WT cells, the autophagosomes may be autophagic membrane structures, demonstrating, at the morphological level, that SARS-CoV-2 uses autophagy to promote its replication. These observations are supported by different studies demonstrating that hydroxychloroquine inhibits SARS-CoV-2 replication by impairing autophagosome–lysosome fusion [61,62]. However, our data cannot exclude the possibility that an antiviral autophagic mechanism can also operate to safeguard the host cells against viral invasion. Alternatively, autophagosomes found in ΔF cells may represent structures used by the cells to supply substrates and energy, a role that cannot be subverted by SARS-CoV-2 in these cells due to the well-known phenomenon of a CF-associated dysregulation of autophagy, due to impaired autophagosome–lysosome fusion and/or degradation [63]. Indeed, CFTR-defective cells are characterized by an increase in the production of reactive oxygen species, leading to an increased activation of transglutaminase-2 that, through inactivating the Beclin1 (BECN1) complex, results in a deficient autophagy process [64] due to sequestration of BECN1 and other interactors into intracellular aggregates [65,66,67]. Moreover, these autophagy-deficient cells showed accumulation of SQSTM1, a protein involved in the formation and elimination of aggregates containing ubiquitinated proteins [68,69]. This SQSTM1 accumulation at the endosomal level reduces GTPase activity of GTPases Rab5 [66] and Rab7 [70], which are essential for maturation of early and late endosomes and phagosomes, respectively. Thus, this autophagy dysfunction reported in CFTR-modified cells could be exploited by intracellular pathogens to survive and persist in eukaryotic cells [71,72].

Our findings showed the presence of lipid droplets exclusively in ΔF cells after SARS-CoV-2 infection, which is an interesting novel finding differentiating CFTR-deficient cells from WT cells. The possibility that the presence of these lipid droplets is in response to SARS-CoV-2 exposure seems likely, in view of the rapidity with which lipids appeared (i.e., 1 hpi). We envisage that lipids play a role in energy supply and synthesis of the cell membrane. In particular, lipophagy, or the lysosomal degradation of lipid droplets [73,74], has been demonstrated to regulate viral replication by enhancing the number of lipid droplets to be used as scaffolds or energy sources [75]. This mechanism has been shown in cells infected with some positive-strand RNA viruses, but it remains little investigated in many viral replication processes, including that of SARS-CoV-2. In our study, we did not observe viral particles within lipid droplets; thus, we speculate that lipophagy does not play a role as a scaffold in the replication of SARS-CoV-2 in infected ΔF cells. Thus, it is likely that the accumulation of lipid droplets in ΔF cells from the earliest times of infection represents the outcome of various cellular processes, such as the need for nutrient supplementation and, thus, energy, or it could be a consequence of a defective autophagic degradation process.

Moreover, since the data in the literature on the distribution of ACE2 are mainly focused on tissue samples and based on immunofluorescence, we performed immune-electron microscopy targeting the ACE2 receptor. Our data revealed that in WT cells at early time points after infection, ACE2 localization is observed at the limiting membrane of cytoplasmic structures, such as tubules, ER vesicles and DMVs, suggesting that ACE2 expression rises in parallel with the increasing complexity of the cytoplasmic membrane remodeling that occurs following SARS-CoV-2 infection.

In late infection, cytoplasmic expression of ACE2 is greatly reduced while membrane expression persists, probably due to the fact that at this stage, replicative structures are reduced while completely formed viral particles appear.

In ΔF cells, ACE2 expression is maintained in cytoplasmic structures, even in the late stages of infection, in contrast to what was observed in WT cells. One possible explanation for this could be that in ΔF cells, membrane reorganization is impaired and it, therefore, takes longer to organize the replicative structures required for viral replication.

## 4. Materials and Methods

### 4.1. Cell Lines and Virus Strain

The human CF bronchial epithelial cell line CFBE41o- [76,77] was derived from a CF patient carrying an F508del^+/+^ CFTR mutation, caused by a deletion of a single phenylalanine at position 508 in the CFTR protein. This cell line was used to obtain WT-CFTR (CFBE41o- WT) or F508del-CFTR (CFBE41o- ΔF) cell lines as previously described [78]. In brief, CFBE41o- cells immortalized with SV40 plasmid (pSVori) were stably transfected with HIV-based lentiviral vector expressing WT-CFTR cDNA or with HIV-based lentiviral vector expressing F508del-CFTR cDNA to generate the ΔF strain [78]. These cell lines were kept in culture in Minimum Essential Medium (MEM, Gibco, Thermo Fisher, Monza, Italy) supplemented with 10% FBS (Euroclone, Pero, Italy) and 1% glutamine (GlutaMAX, Gibco, Thermo Fisher) and grown to a monolayer after passaging. To maintain the selection of WT- and ΔF-CFTR-overexpressing subclones, puromycin (Gibco, Thermo Fisher) was added to the growth medium during cell culture as selective agent (0.5 mg/mL for CFBE41o- WT and 2 mg/mL for CFBE41o- ΔF cells).

The SARS-CoV-2 strain used (GR-SARS-CoV-2) was isolated from respiratory secretions from a COVID-19-positive adult male at S. Orsola Hospital (Bologna, Italy) in March 2020 and replicated as described by Ogando and colleagues [79].

### 4.2. Cell Culture and Infection

WT and ΔF CFBE41o- cells were cultured in adhesion in a 12-well plate. Once approximately 70% confluence was reached, cells were inoculated with SARS-CoV-2 at a multiplicity of infection (MOI) of 1 and incubated at 37 °C and 5% CO_2_ for 1 h. Next, the inoculum was removed, the cells were washed twice with PBS and fresh medium was added. After 1, 3, 6, 24, 48 and 72 h post infection (hpi), the media were discarded, the cells were washed twice with PBS and cell pellets were harvested. Non-infected cultured cells served as controls.

### 4.3. Transmission Electron Microscopy (TEM)

For ultrastructural examination, cell pellets of WT and ΔF CFBE41o- cells that were mock infected or infected with SARS-CoV-2 were fixed for 1 h in 2% glutaraldehyde in 0.1 M phosphate buffer (PB) at different hpi and, after washes, postfixed for 1 h in 1% OsO_4_ diluted in 0.2 M K_3_Fe(CN)_6_. After rinsing in 0.1 M PB, the samples were dehydrated in graded concentrations of acetone and embedded in a mixture of Epon and Araldite (Electron Microscopic Sciences, Fort Washington, PA, USA). Ultrathin sections were cut at 70 nm thickness on an Ultracut E ultramicrotome (Reichert-Jung, Heidelberg, Germany), contrasted with lead citrate and observed on a Philips Morgagni 268 D electron microscope (Fei Company, Eindhoven, The Netherlands), equipped with a Megaview II camera for acquisition of digital images. To better understand the morphology of the membranes that were remodeled due to SARS-CoV-2 infection, additional samples were fixed with 2% glutaraldehyde containing 0.5% tannic acid and then processed as described above. Tannic acid is a good fixative of peptide and proteins and is useful for increasing the electron density in biological samples [80]. The technique was applied on WT and ΔF CFBE41o- cells at baseline and at 24, 48 and 72 hpi.

### 4.4. Immunoelectron Microscopy

To detect the expression of ACE2, immunohistochemical staining was carried out on uninfected and infected WT and ΔF CFBE41o- cells at 24, 48 and 72 hpi using the avidin-biotin–peroxidase complex (ABC) technique. Samples were fixed in 4% paraformaldehyde in 0.1 M PB, pH 7.4, for 2 h at 4 °C and subsequently incubated in a solution of 3% hydrogen peroxide in H_2_O for 30 min to quench the endogenous peroxidase. After washing in 0.05 M PB, pH 7.6, samples were treated with 2% normal goat serum for 1 h and then the polyclonal antibody anti-ACE2 (abcam ab15348, Prodotti Gianni, Milan, Italy)diluted 1:200, was applied overnight. After three washes with Tris-HCl buffer, cells were reacted with biotinylated goat anti-rabbit immunoglobulins (DAKO, Milan, Italy), diluted 1:400, for 1 h. The immunoreaction was detected using a Vectastain Elite ABC kit (Vector, Burlingame, CA, USA) and then visualized by 3,3 diaminobenzidine (DAB) tetrahydrochloride (Dako, Milan, Italy) for 5–10 min. The control for the specificity of the immunoreaction was established by omitting the primary antibody. No electron-dense deposit was observed on negative control samples. Subsequently, the samples were postfixed in 1% OsO_4_ in PB for 1 h and then processed for TEM examination as described above. The samples were examined without additional post staining. The immune-electron pattern was considered positive when a black, electron-opaque, DAB deposit was observed on the cell structures in contrast with the background.

## 5. Conclusions

In this study, we analyzed the ultrastructural changes induced by SARS-CoV-2 infection in WT and ΔF CFBE41o- cells at various time points after infection. Our data are mainly based on morphological criteria, which is a limitation of the work.

Our ultrastructural data suggest that SARS-CoV-2 can enter and infect both WT and ΔF cells, but CFTR-modified cells have difficulties in organizing normo-conformed replication complexes leading to poor, inefficient and aberrant viral particle formation.

In addition to being expressed at the plasma membrane, ACE2 is also present in the cytoplasm at early time points in WT cells, while in ΔF cells, it persists in the cytoplasm, even at late times of infection.

We do not know if our data can be directly or indirectly related to the development of milder symptoms in cases of COVID-19 described in pwCF. It is possible that their response, as already suggested, may depend on characteristics of the mucus, prudent behavior of patients or ongoing therapies. However, since the mechanisms by which viral infections are affected by the respiratory system impairment in pwCF are poorly understood, we believe that the present study may be useful to characterize the impaired process of production and secretion of viral particles in CFTR-modified airway cells.

This study presents a novel description of the SARS-CoV-2 cell cycle in human bronchial epithelial cells, with and without CFTR-modification, and could pave the way to a better understanding of the impact of COVID-19 on pwCF by characterizing SARS-CoV-2 infection in ΔF-mutated CFTR cells.

## Figures and Tables

**Figure 1 ijms-23-09724-f001:**
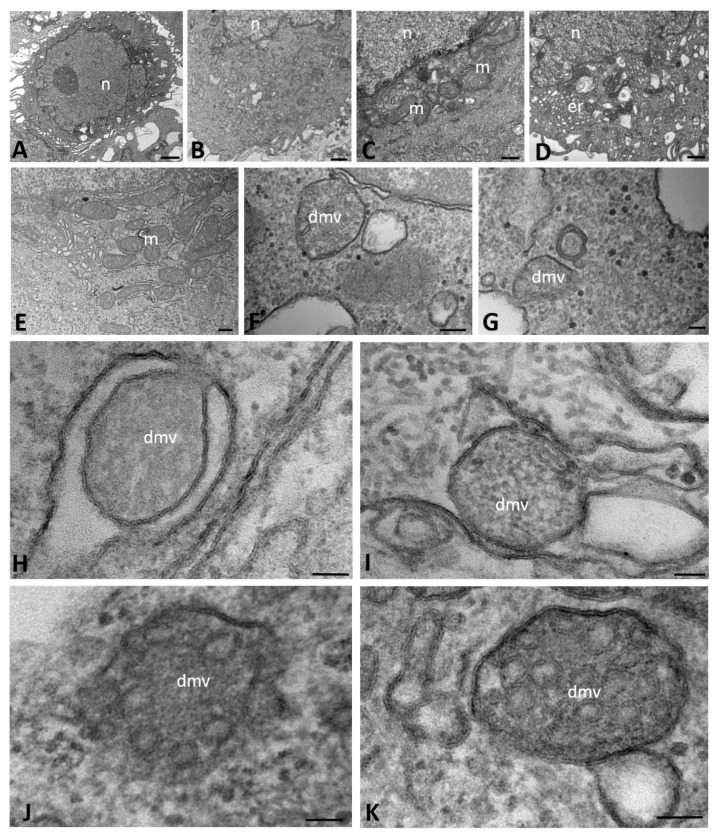
Transmission electron microscopy showing the morphology of mock-infected (**A**–**D**) and SARS-CoV-2-infected (**E**–**K**) CFBE41o- WT cells at early time points of infection (from 1 to 6 h post infection). Double-membrane vesicles with granular (**F**–**I**) and vesicular (**J**,**K**) content are visible. (dmv): double-membrane vesicles; (er): endoplasmic reticulum; (m): mitochondria; (n): nucleus. Bars: (**A**) 1 µm; (**B**,**D**) 500 nm; (**C**,**E**) 200 nm; (**F**) 100 nm; (**G**–**K**) 50 nm.

**Figure 2 ijms-23-09724-f002:**
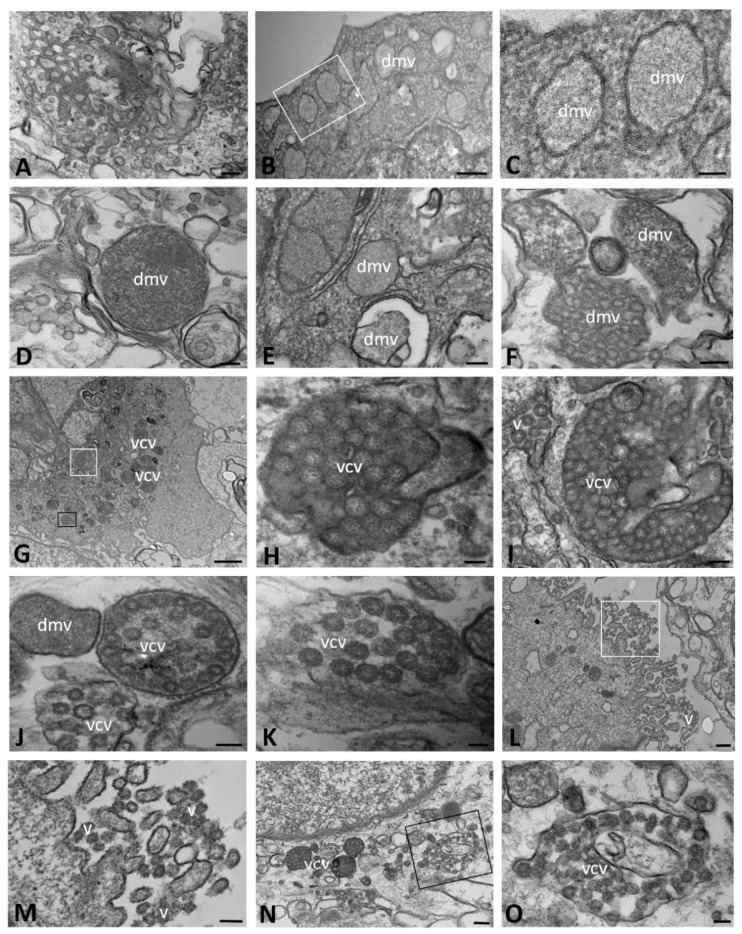
Transmission electron microscopy showing the viral replicative structures in SARS-CoV-2-infected CFBE41o- WT cells at late time points of infection (from 24 to 72 h post infection). Double-membrane vesicles with granular (**B**,**C**,**E**,**F**) and vesicular (**D**,**F**) content are visible. Virions are detected in single-membrane-bound vesicles (**J**–**K**,**N**,**O**) or on the cell surface (**L**,**M**). In (**B**–**D**), the sample was treated with tannic acid. The boxed area in (**B**,**L**,**N**) is enlarged in (**C**,**M**,**O**), respectively. The black boxed area in (**G**) is enlarged in (**H**) and the white boxed area is enlarged in (**I**). (dmv): double-membrane vesicle; (v): virions; (vcv): virion-containing vesicle. Bars: (**A**,**D**–**F**,**I**,**J**,**M**,**O**) 100 nm; (**B**,**L**,**N**) 200 nm; (**C**,**H**,**K**) 50 nm; (**G**) 1 µm.

**Figure 3 ijms-23-09724-f003:**
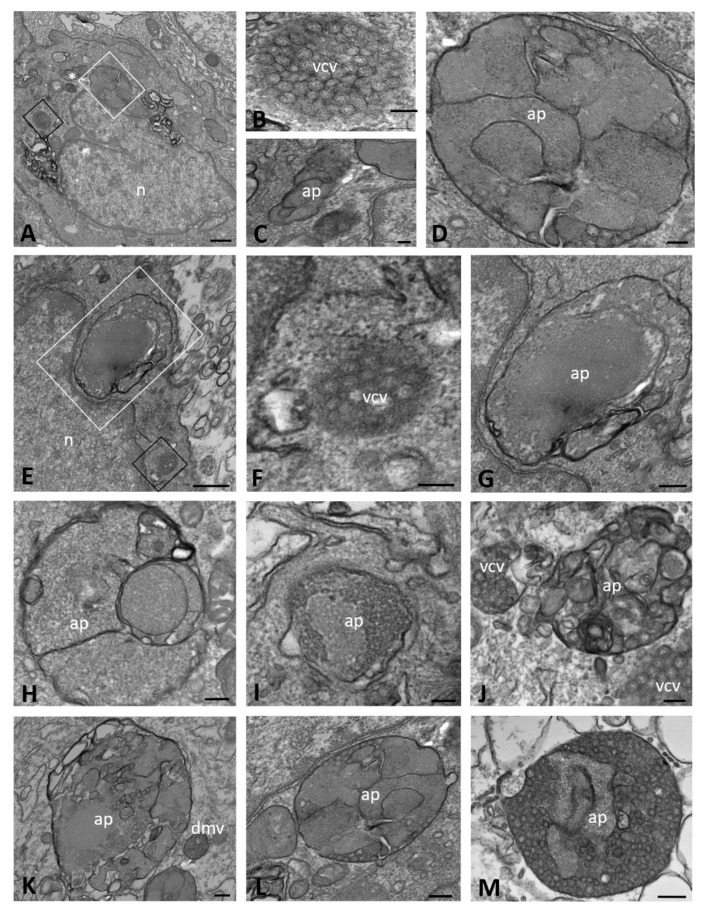
Transmission electron microscopy showing autophagosome structures at late time points of SARS-CoV-2 infection (from 24 to 72 h post infection) in CFBE41o- WT cells. Autophagosomes with a granular (**D**,**G**–**J**) or mixed content with evident viral particles (**D**,**K**–**M**) are detected. The black boxed area in (**A**) is enlarged in (**B**), the white boxed area is enlarged in (**D**) and the asterisk is enlarged in (**C**). The black boxed area in (**E**) is enlarged in (**F**) and the white boxed area is enlarged in (**G**). (ap): autophagosome; (n): nucleus; (vcv): virion-containing vesicle. Bars: (**A**,**E**) 500 nm; (**G**,**H**,**J**,**K**,**M**) 200 nm; (**B**–**D**,**F**,**I**,**L**) 100 nm.

**Figure 4 ijms-23-09724-f004:**
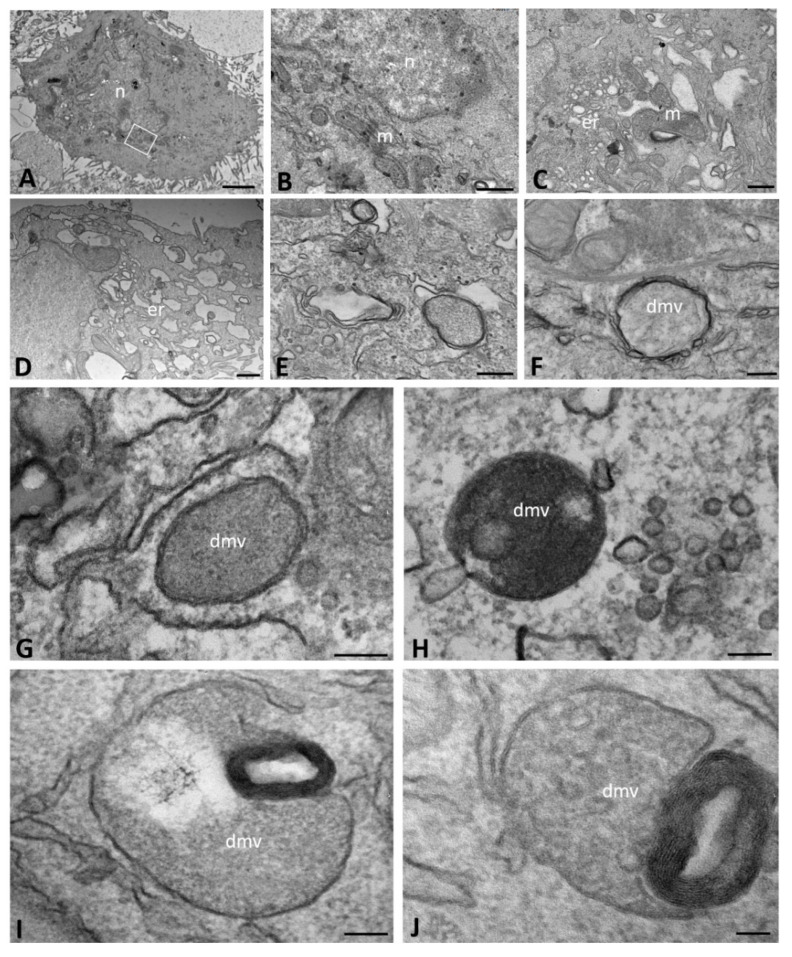
Transmission electron microscopy showing the morphology of mock-infected (**A**–**C**) and SARS-CoV-2-infected (**D**–**J**) CFBE41o- ΔF cells at early time points of infection (from 1 to 6 h post infection). Double-membrane vesicles with granular (**E**–**I**) and vesicular (**J**) content are visible. The boxed area in (**A**) is enlarged in (**B**). (dmv): double-membrane vesicle; (m): mitochondria; (n): nucleus. Bars: (**A**) 2 µm; (**B**,**C**) 500 nm; (**E**,**F**) 200 nm; (**G**–**I**) 100 nm; (**J**) 50 nm.

**Figure 5 ijms-23-09724-f005:**
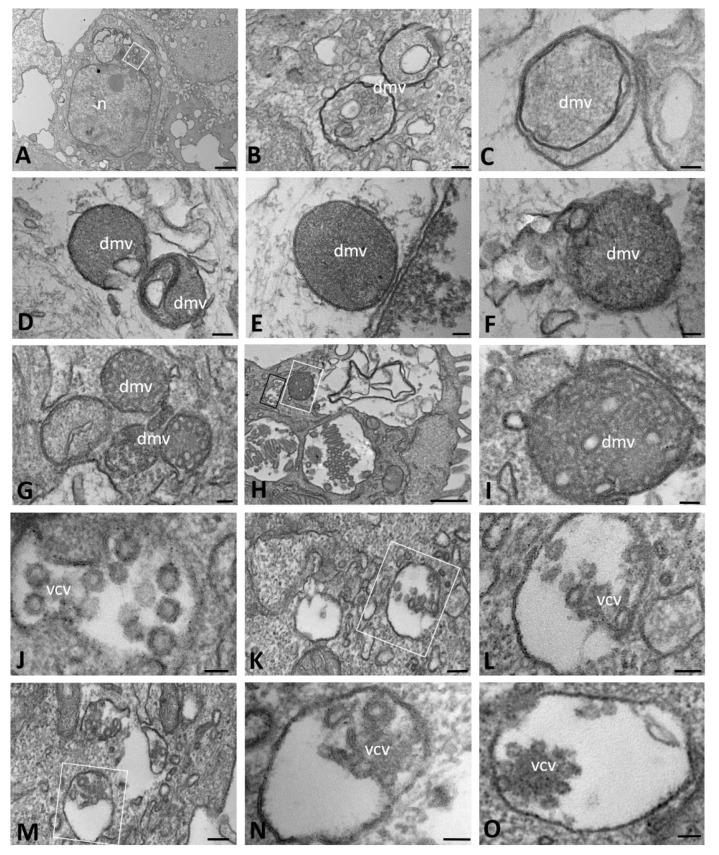
Transmission electron microscopy showing viral replicative structure of SARS-CoV-2 infected CFBE41o- ΔF cells at late time points of infection (from 24 to 72 h post infection). Double-membrane vesicles with granular (**A**–**F**) and vesicular (**G**–**I**) content are visible. Virions are detected in single-membrane-bound vesicles (**J**–**O**) lacking well-defined organization. The boxed areas in (**A**,**K**,**M**) are enlarged in (**B**,**L**,**N**), respectively. The black boxed area in (**H**) is enlarged in (**J**) and the white boxed area is enlarged in (**I**). (dmv): double-membrane vesicle; (n): nucleus; (vcv): virion-containing vesicle. Bars: (**A**) 1 µm; (**H**) 500 nm; (**B**,**D**,**K**,**M**) 100 nm; (**C**,**E**–**G**,**I**,**J**,**L**,**N**,**O**) 50 nm.

**Figure 6 ijms-23-09724-f006:**
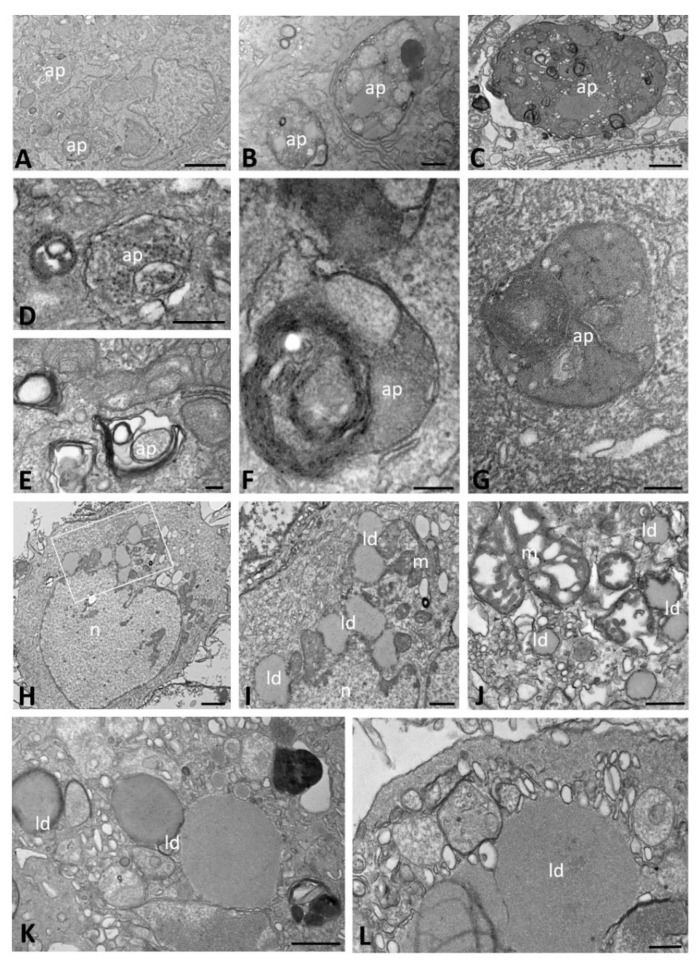
Transmission electron microscopy showing autophagosome structures (**A**–**G**) and lipid droplets (**H**–**L**) in SARS-CoV-2-infected CFBE41o- ΔF cells. Autophagosomes are visible at early (**A**–**C**) and late times of infection (**D**–**G**). In (**G**), the sample was treated with tannic acid. The boxed area in (**H**) is enlarged in (**I**). (ap): autophagosome; (ld): lipid droplet; (m): mitochondria; (n): nucleus. Bars: (**A**) 2 µm; (**C**,**H**) 1 µm; (**B**,**I**–**K**) 500 nm; (**D**,**L**) 200 nm; (**E**–**G**) 100 nm.

**Figure 7 ijms-23-09724-f007:**
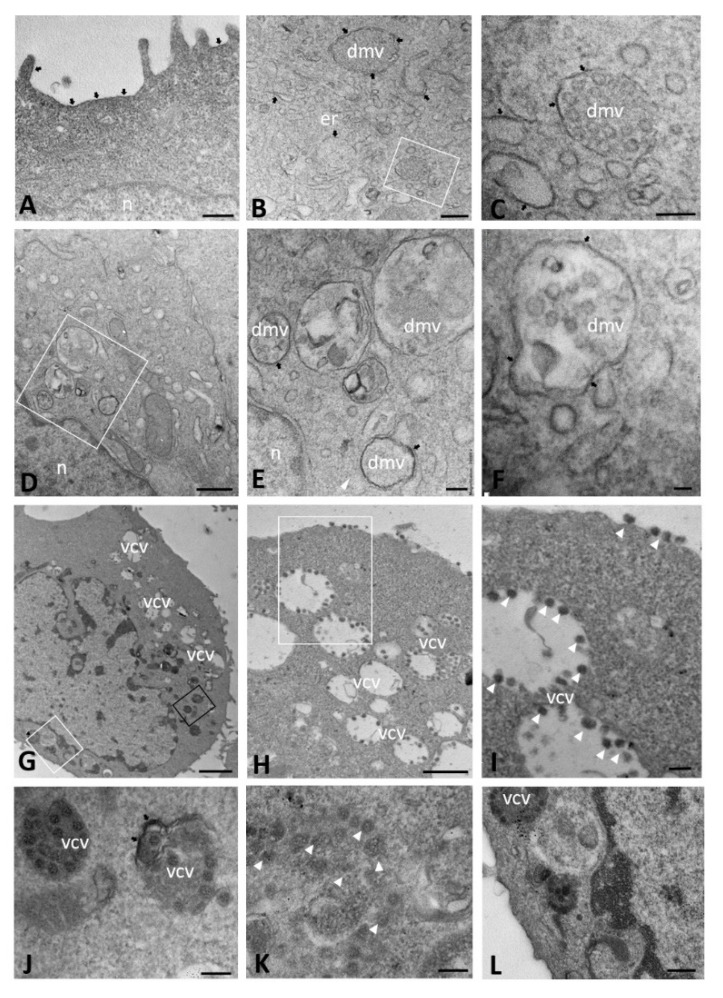
Immuno-electron microscopy showing ACE2 localization in mock-infected (**A**) and SARS-CoV-2-infected CFBE41o- WT cells at 24 (**B**,**C**), 48 (**D**–**F**) and 72 (**G**–**L**) hours post infection. The boxed areas in (**B**,**D**,**H**) are enlarged in (**C**,**E**,**I**), respectively. In (**G**), the black boxed area is enlarged in (**J**) and the white boxed area is enlarged in (**L**). (dmv): double-membrane vesicle; (n): nucleus; (vcv): virion-containing vesicle. Arrowheads indicate virions. Arrows demarcate ACE2 immunoreactivity. Bars: (**G**) 1 µm; (**D**,**H**) 500 nm; (**A**,**B**,**L**) 200 nm; (**C**,**E**,**I**–**K**) 100 nm; (**F**) 50 nm.

**Figure 8 ijms-23-09724-f008:**
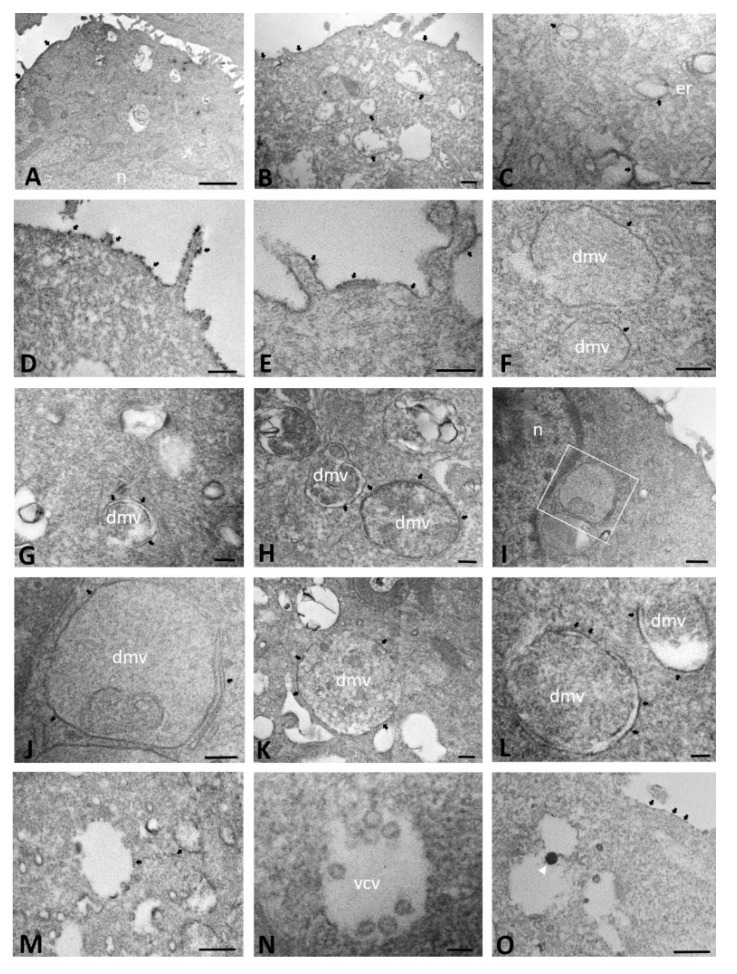
Immuno-electron microscopy showing ACE2 localization in mock-infected (**A**) and SARS-CoV-2-infected CFBE41o- ΔF cells at 24 (**B**–**F**), 48 (**G**–**L**) and 72 (**M**–**O**) hours post infection. The boxed area in (**I**) is enlarged in (**J**). (dmv): double-membrane vesicles; (er): endoplasmic reticulum; (n): nucleus; (vcv): virions-containing vesicle. Arrowhead indicates virion. Arrows demarcate ACE2 immunoreactivity. Bars: (**A**) 1 µm; (**B**,**D**,**H**,**I**,**K**,**M**,**O**) 200 nm; (**C**,**E**–**G**,**J**,**L**,**N**) 100 nm.

**Table 1 ijms-23-09724-t001:** Description of main morphological characteristics in SARS-CoV-2-infected CFBE41o- WT and ΔF cells in early and late stages of infection.

Time of Infection	CFBE41o- Cells	DMVs		Autophagosomes		Lipid Droplets	Proteinaceous Material	Virions		ACE2 Expression	
		Granular Content	Vesicle Structures	Cellular Material	Replicative Structure			Cytoplasmic Vesicles	Cell Membrane	Cytoplasmic Membrane	Cell Membrane
Early	WT	●	●	-	-	-	●	-	-	●	●
	ΔF	●	-	●	-	●	●	-	-	●	●
Late	WT	●	●	-	●	-	●	●	●	-	●
	ΔF	●	●	●	-	●	-	●	-	●	●

●: present; -: not present; DMVs: double-membrane vesicles.

## Data Availability

Not applicable.

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
