# Peer review of "Ultrastructural Characterization of Human Bronchial Epithelial Cells during SARS-CoV-2 Infection: Morphological Comparison of Wild-Type and CFTR-Modified Cells"

_ijms, 2022, doi:10.3390/ijms23179724_

Round 1
Reviewer 1 Report
The authors analyzed the morphological impact of CFTR mutation in human bronchial epithelial cells and contrasted to wild-type cells over 72-hour period post infection with SARS-CoV-2 or mock. Comparison of early and late time points revealed important differences in ACE2 expression, autophagosomes and in double membrane vesicles in terms of shape and viral particles. Overall, this article presents novel description of ultrastructural changes during SARS-CoV-2 cell cycle in the context of CFTR membrane protein mutation background. However, it would be great to address some concerns as below:
1. Line 148-150 mentions Fig. 2L-M and Fig. 2N-O. However, Figure2 has sub-figure parts upto ‘K’ only.
2. Line 167, mentions Fig. 3A-D and 3E-G. It could be simplified as Fig. 3A-G, unless authors meant Fig. 3A,D and 3E,G or (Fig. 3A-D and 3E-G respectively) for referring two different sub-parts in the sentence.
3. It would be good if authors could briefly explain “immunoreactivity” in the text and how its localization could be identified in the images generated by microscopy from the rest.
4. Having statistics done based on microscopy images is crucial to conclude the claims by authors in Table1 about the morphological changes in WT compared to cell line with mutated CFTR (for example, P < 0.05 or not). Having this analysis done will make paper much stronger and results are likely to be significant.
5. Line 433 has a typo, “thst”.
6. Claims of the paper could become much stronger and relevant if another experiment with over-expression of wild-type CFTR is done in cells with CFTR mutation to test if double membrane vesicles regain right shape and number of viral particles in them along with presence of autophagosomes at late stage of infection. This will confirm the role of CFTR in SARS-CoV-2 infection.
Author Response
- Line 148-150 mentions Fig. 2L-M and Fig. 2N-O. However, Figure2 has sub-figure parts up to ‘K’ only.
Many thanks. We made a mistake in adding the figures in the manuscript. We have uploaded Figure 1 for both Figure 1 and Figure 2. Please, find in the revised manuscript the correct Figure 2.
- Line 167, mentions Fig. 3A-D and 3E-G. It could be simplified as Fig. 3A-G, unless authors meant Fig. 3A,D and 3E,G or (Fig. 3A-D and 3E-G respectively) for referring two different sub-parts in the sentence.
Thank you. We have simplified the sentence (Fig. 3A-G) as suggested.
- It would be good if authors could briefly explain “immunoreactivity” in the text and how its localization could be identified in the images generated by microscopy from the rest.
Thanks for requesting this clarification. In the results sections 2.3 and 2.4 we added a clarifying sentence, at line 264-265 and 292-293 respectively.
- Having statistics done based on microscopy images is crucial to conclude the claims by authors in Table1 about the morphological changes in WT compared to cell line with mutated CFTR (for example, P < 0.05 or not). Having this analysis done will make paper much stronger and results are likely to be significant.
Many thanks for the interesting suggestion. However, the table we presented in the manuscript only summarize qualitative information indicating presence or absence of the characteristics we have evaluated, which makes statistical analysis impossible. As we pointed out along the manuscript, our data are mainly based on morphological criteria, which is a limitation of the work.
- Line 433 has a typo, “thst”.
We corrected the typo. The correct word was “that”.
- Claims of the paper could become much stronger and relevant if another experiment with over-expression of wild-type CFTR is done in cells with CFTR mutation to test if double membrane vesicles regain right shape and number of viral particles in them along with presence of autophagosomes at late stage of infection. This will confirm the role of CFTR in SARS-CoV-2 infection.
Many thanks for the very interesting suggestion and comment. The CFBE41o- wild-type cells we have used in this study derived from a CF patient carrying the F508del mutation in homozygosis which were converted to wild-type CFTR expressing cells thanks to the insertion of the wt-CFTR plasmid (for more details, chapter “Cell lines and virus strain”, materials and methods section). Thus, we can consider these cells as CFTR mutated genetically converted to wild-type expressing CFTR. Another strategy to confirm the role of CFTR in SARS-CoV-2 infection could be using available CFTR modulators (e.g. Trikafta, Ivacaftor) on CFTR mutated cells to rescue the CFTR function, condition that we have already tested by molecular analysis (Lotti et al. Cells 2022). We are already working to answer this very intriguing question, but it will require a long time and a specific paper to explain what we will find. Anyway, is to be emphasized that the possible CFTR rescue of CFTR mutated cells does not reach more than 10-20%, which could not be sufficient to bring the expected morphological changes.
Reviewer 1 also indicate “Moderate English changes required”. We had the English revised by “Cambridge proofreading and editing LCC” before submission. Please find the certificate in the document uploaded.

Reviewer 2 Report
In this manuscript, Merigo et al. provided an interesting ultrastructural evaluation of bronchial cells, comparing Sars-CoV-2 infected cells and a mocked-infected control group in both wild-type cells and in a model of cystic fibrosis. They compared ultrastructural features at different time points and evaluated the ACE2 immunoelectron microscopic localization, providing an admirable set of images and figures.
Overall, the manuscript is well written and the approach is systematic and stepwise.
1) Please, consider providing more details regarding how CFTR could regulate Sars-CoV-2 replication (Introduction, page 2 lines 82-85)
2) Consider adding more references, particularly:
I) in the Introduction when describing the different Coronavirus types and clinical manifestations (lines 40-44, page 1);
II) in the Introduction when reporting that “A recent study demonstrated that SARS-CoV-2 replication is reduced in bronchial cells from people with cystic fibrosis (pwCF), a lethal genetic disease with autosomal recessive inheritance caused by mutations in CFTR gene [21], which codes for the CFTR membrane protein, an ion channel involved in both chloride and bicarbonate ion transport [22-25].” (page, lines 75-78), please consider adding the study you are referring to at the beginning of the sentence: references [21-25] are published in the 90’-early 2000 and none of them seems to refer to the evaluation of SARS-CoV-2 replication in bronchial cells.
III) in the Discussion when reporting that “Previous analyses of ultrathin sections using TEM described how coronaviruses use host cell membranes to form specialized structures such as DMVs and CM during their replication cycle, a process already described in a wide range of cultured cells or tissues” (lines 316-318, page 15);
3) Provide the extended nomenclature of the “TMPRSS-2” protease, before the acronym (line 53, page 2);
4) The page count is altered after the landscape-orientated page (page 14);
5) Page 7, line 193; consider reporting that, starting from this line, the cells described are Sars-CoV-2 infected.
Author Response
1) Please, consider providing more details regarding how CFTR could regulate Sars-CoV-2 replication (Introduction, page 2 lines 82-85).
We added a possible explanation at lines 89-90. However, we prefer not to add too much information in the introduction since the mechanism behind the involvement of CFTR in the regulation of SARS-CoV-2 replication is still to be better elucidated.
2) Consider adding more references, particularly:
- I) in the Introduction when describing the different Coronavirus types and clinical manifestations (lines 40-44, page 1);
We added references from 1 to 13.
- II) in the Introduction when reporting that “A recent study demonstrated that SARS-CoV-2 replication is reduced in bronchial cells from people with cystic fibrosis (pwCF), a lethal genetic disease with autosomal recessive inheritance caused by mutations in CFTR gene [21], which codes for the CFTR membrane protein, an ion channel involved in both chloride and bicarbonate ion transport [22-25].” (page, lines 75-78), please consider adding the study you are referring to at the beginning of the sentence: references [21-25] are published in the 90’-early 2000 and none of them seems to refer to the evaluation of SARS-CoV-2 replication in bronchial cells.
We added the reference of the study reporting the reduction of SARS-CoV-2 infection in cells derived from pwCF (ref 34) at the beginning of the sentence. However, the other references (in the original manuscript 21-25, in the revised one 35-39) were pertinent since they referred to the description of the cystic fibrosis disease.
III) in the Discussion when reporting that “Previous analyses of ultrathin sections using TEM described how coronaviruses use host cell membranes to form specialized structures such as DMVs and CM during their replication cycle, a process already described in a wide range of cultured cells or tissues” (lines 316-318, page 15);
We added reference 45.
3) Provide the extended nomenclature of the “TMPRSS-2” protease, before the acronym (line 53, page 2);
We added the extended nomenclature of TMPRSS2 at line 56.
4) The page count is altered after the landscape-orientated page (page 14);
Many thanks. We have solved the problem.
5) Page 7, line 193; consider reporting that, starting from this line, the cells described are Sars-CoV-2 infected.
Many thanks. As requested, we added at line 197 that the cells described are SARS-CoV-2 infected.

Round 2
Reviewer 1 Report
Thanks for addressing the comments and incorporating the suggestions!